# Hepatocellular Carcinoma Cell-Derived Exosomal miR-21-5p Induces Macrophage M2 Polarization by Targeting RhoB

**DOI:** 10.3390/ijms24054593

**Published:** 2023-02-27

**Authors:** Haiyang Yu, Jing Pan, Siyue Zheng, Deyang Cai, Aixiang Luo, Zanxian Xia, Jufang Huang

**Affiliations:** 1Department of Anatomy and Neurobiology, School of Basic Medical Sciences, Central South University, Changsha 410083, China; 2Department of Bioinformatics, School of Life Sciences, Central South University, Changsha 410083, China; 3Institute of Biopharmaceutical and Health Engineering, Tsinghua Shenzhen International Graduate School, Tsinghua University, Shenzhen 518055, China; 4Department of Cell Biology, School of Life Sciences, Central South University, Changsha 410083, China; 5Hunan Key Laboratory of Animal Models for Human Diseases, School of Life Sciences, Central South University, Changsha 410083, China; 6Hunan Key Laboratory of Medical Genetics & Center for Medical Genetics, School of Life Sciences, Central South University, Changsha 410083, China

**Keywords:** HCC-derived exosomes, miR-21-5p, macrophages, polarization, RhoB

## Abstract

M2-like polarized tumor-associated macrophages (TAMs) are the major component of infiltrating immune cells in hepatocellular carcinoma (HCC), which have been proved to exhibit significant immunosuppressive and pro-tumoral effects. However, the underlying mechanism of the tumor microenvironment (TME) educating TAMs to express M2-like phenotypes is still not fully understood. Here, we report that HCC-derived exosomes are involved in intercellular communications and exhibit a greater capacity to mediate TAMs’ phenotypic differentiation. In our study, HCC cell-derived exosomes were collected and used to treat THP-1 cells in vitro. Quantitative polymerase chain reaction (qPCR) results showed that the exosomes significantly promoted THP-1 macrophages to differentiate into M2-like macrophages, which have a high production of transforming growth factor-β (TGF-β) and interleukin (IL)-10. The analysis of bioinformatics indicated that exosomal miR-21-5p is closely related to TAM differentiation and is associated with unfavorable prognosis in HCC. Overexpressing miR-21-5p in human monocyte-derived leukemia (THP-1) cells induced down-regulation of IL-1β levels; however, it enhanced production of IL-10 and promoted the malignant growth of HCC cells in vitro. A reporter assay confirmed that miR-21-5p directly targeted Ras homolog family member B (RhoB) 3′-untranslatedregion (UTR) in THP-1 cells. Downregulated RhoB levels in THP-1 cells would weaken mitogen-activated protein kinase (MAPK) axis signaling pathways. Taken together, tumor-derived miR-21-5p promote the malignant advance of HCC, which mediated intercellular crosstalk between tumor cells and macrophages. Targeting M2-like TAMs and intercepting their associated signaling pathways would provide potentially specific and novel therapeutic approaches for HCC treatment.

## 1. Introduction

HCC is the most common form of primary liver cancer, with rapid progression, poor prognosis, and a high risk of resistance and recurrence [1]. Although great advances have been made in surgical resection, liver transplantation, radiofrequency ablation, and other non-surgical interventions, the prognosis of most HCC patients remains unsatisfactory, with a 5-year survival rate of approximately 20% [2]. Thus, new therapeutic approaches for HCC are still in urgent need.

In recent years, immunotherapy has exhibited a great potential in treating malignant tumors. It is considered an effective and safe approach for precision medicine-based treatment of HCC. However, only a small portion of patients benefit from these anti-HCC therapies, which are accompanied by resistance and several immune-related adverse events [3]. For large, solid tumors, it is hard for effector lymphocytes to enter the tight spaces between tumor tissues. Moreover, TME can weaken the effect of effector lymphocytes by secreting a series of inhibitors. TME is a complex assembly of a variety of cell types, including macrophages and monocytes, among the tumor-associated myeloid cell infiltrates, which mediate the function of immunotherapy [2].

Macrophages are an important part of innate immunity and a key factor of adaptive immunity initiation. Compared with normal macrophages, TAMs are imperfectly differentiated macrophages prone to immunosuppressive phenotypes [4]. Under the effect of multi-factors, macrophages are recruited into tumor tissue and educated to be important accomplices in promoting malignant progression of tumor. Various cytokines in TME are extremely important for the recruitment of macrophages. Chemokine axes such as C-C motif chemokine ligand 2 (CCL-2)/C-C motif chemokine receptor 2 (CCR2) and CCL-5/CCR5 can recruit monocytes/macrophages from the blood to the tumor region. IL-10 and macrophage colony stimulating factor (M-CSF) secreted by tumor cells can also promote the migration of macrophages to tumor sites [5]. Furthermore, in the latest reports, exosomes are thought to be involved in this process, which has been demonstrated in a variety of tumors [6,7,8].

Exosomes, measuring from 30 to 150 nm in diameter, are microvesicles formed in multivesicular bodies, which release exosomes into the extracellular milieu by fusing with cytomembranes [9,10]. Exosomes can be produced by various types of cells and serve as mediators in intercellular communications by transporting information cargo, such as proteins, lipids, and nucleic acids [11]. Specific proteins highly enriched in exosomes, such as Tsg101, CD63, Hsp70, CD9, and CD81, are usually used as markers to identify exosomes [12]. Numerous research reports have pointed out that exosomes mediate regulation of the phenotypes of TAMs in TME [6,7,8]. 

In our study, we investigated the effect of tumor cell-derived exosomes on macrophages and identified the critical tumor-derived exosomal miRNA’s function in the polarization of M2-like macrophages on the basis of an analysis of databases and in vitro studies. It offers new opportunities for potential therapeutic strategies of targeting TAM in HCC.

## 2. Results

### 2.1. Increased M2-like TAM Infiltration Is Associated with Poor Patient Prognosis in HCC

TAMs are a highly heterogeneous population with different functions in HCC, which poses the challenge to the therapeutic approaches targeting TAMs. The emergence of single-cell RNA sequencing (scRNA-seq) provides us with a powerful tool for the investigation of TAM subtypes and their interaction in TME. We utilized the online tool at “http://cancer-pku.cn:3838/HCC/ (accessed on 25 August 2022)”, provided by Zemin Zhang lab at Peking University, to identify the biomarkers of macrophage subpopulation, which are specifically distributed in normal or tumor tissues of HCC. Referring to the analysis of datasets, we identified three representative marker genes of a subpopulation of TAMs in HCC, including CD68, CD5L, and mannose receptor C-type 1 (MRC1). The heatmaps of marker expression on the reduced dimensions showed that CD68 is the classical biomarker that is highly expressed by a large majority of macrophages; however, CD5L and MRC1 are differentially distributed with specific expression in adjacent non-malignant or tumor tissue (Figure 1A,B). CD5L is a secreted protein mainly expressed by macrophages in lymphoid and inflamed tissues [13]. It is considered an innate immune effector response to foreign infection [14]. MRC1 is recognized as a marker of M2-like polarized macrophages, which are highly expressed in TAMs in HCC [15,16]. Further, MRC1+ macrophages are much more abundant in tumor than in normal tissues, which is consistent with previous studies [17].

We then used the publicly available dataset from the Tissue and Pathology sections of the Human Protein Atlas “https://www.proteinatlas.org/ (accessed on 25 August 2022)”to analyze the correlation between the infiltration of TAM subtypes and the prognosis of patients. In most tumor tissue, we observed high level CD68^+^ and MCR1^+^ macrophage infiltration, and both of these are associated with an unfavorable prognosis of HCC. In some of the HCC tissues, we found much CD5L^+^ macrophage infiltration; however, the density of CD5L-positive macrophages is normally associated with a better prognosis (Figure 1C). Therefore, we believe that the accumulation of M2-like macrophages in HCC is most likely the efficient factor for the malignant progression of tumors.

### 2.2. Tumor-Derived Exosomes Regulate Macrophage Polarization

In light of the fact that exosome secretion is an important way for tumors to shape the TME, it rationally leads one to question whether exosomes participate in promoting the changes of macrophage polarity. We isolated and purified exosomes from conditioned media of tumor cell lines (HepG2 and Huh7) through the standard exosome isolation method of ultrafiltration concentration and precipitation. The shapes of the structures and size distributions of the isolated exosomes were identified using electron microscopy and a Mastersizer (Figure 2A,C). In addition, the detection of characteristic Hsp70 and TSG101 further verified that the isolated particles were exosomes (Figure 2B). In in vitro study, we treated the THP-1 cells with phorbol 12-myristate 13-acetate (PMA) to differentiate into macrophages (Figure 2D,E) and co-cultured these with tumor-derived exosomes interferon-γ (IFN-γ)/lipopolysaccharide (LPS) or IL-4. qPCR results indicated that the effect of tumor-derived exosomes is very similar to that of IL-4, which efficiently increased the immunosuppressive factors, including TGF-β1 and IL-10 (Figure 2F). Moreover, we also observed exosomal dependence of associated cytokine production. It suggests that tumor-derived exosomes are potential effectors of promoting M2-like polarization of macrophages in TME.

### 2.3. Tumor-Derived Exosomal miRNA-21-5p Mediates Macrophage Differentiation into M2-like Polarization

To understand how tumor-derived exosomes regulate macrophage polarization by their cargoes, we analyzed associated data from The Gene Expression Omnibus (GEO) and The Cancer Genome Atlas (TCGA) datasets. Firstly, we screened the top 30 highly expressed miRNAs from the database (GSE106452) by heatmap, in which all of the exosomal miRNA are derived from the culture supernatants of HCC cell lines (CSQT-2, HCC-LM3, HepG2, and MHCC-97L) (Figure 3A). We then analyzed the differentially expressed miRNAs in tumor and adjacent tissue in the TCGA Liver Hepatocellular Carcinoma (LIHC) dataset (Figure 3B). Combining the result sets of two queries by intersection, we found two potential efficient factors in the induction of macrophage polarization: mir-21-5p and mir-1915-3p (Figure 3C). According to the TCGA LIHC dataset, there is a proof that mir-21-5p is highly expressed in HCC tumor tissue with a worse prognosis (Figure 3D,E). Therefore, we hypothesized that mir-21-5p probably plays a key role in regulating macrophage M2-like polarization.

To further verify the prediction, we treated THP-1 cells with Huh-7 exosomes that were labeled with PKH67. We observed that tumor-derived exosomes accumulated in macrophages, and the levels of mir-21-5p were significantly upregulated (Figure 4A,B). To understand the effects of mir-21-5p in macrophages, we transfected mir-21-5p mimics into THP-1 cells, and a supernatant was used to co-culture with hepatic tumor cells. The results indicated that overexpressed mir-21-5p can induce the production of IL-10 in a high level and promote the proliferation of tumor cells (Figure 4C,D), consistent with our previous studies.

### 2.4. Tumor-Derived miR-21-5p Directly Targets RhoB in Regulating Macrophage Polarization

In order to identify the target of miR-21-5p in M2-like polarization of TAMs in HCC, we made a prediction using bioinformatics tools. The expression data of genes were downloaded from TCGA LIHC project. We used a Pearson correlation coefficient analysis to identify associated genes that are highly correlated with miR-21-5p in HCC. These genes were further confirmed by calculating the specific binding sequence of associated genes with miRanda. We then intersected the results with the data that we retrieved from starBase 3.0 “https://starbase.sysu.edu.cn/index.php (accessed on 6 September 2022)”. There are nine potential target genes we obtained, including CPEB3, NIPAL1, RhoB, PPP1R3B, CFL2, SLC39A14, KLHL15, SLC31A1, and KLF9 (Figure 5A).

On the other hand, we performed analysis of differently expressed genes (DEGs) between M1 and M2 subtypes of macrophage groups in two publicly available datasets (GSE66805, GSE95405). A total of 204 DEGs were identified in both of the datasets, including 110 upregulated genes and 94 downregulated genes (Figure 5B). Gene ontology (GO) enrichment analysis showed that most of these differentially expressed genes were related to inflammation and immune responses (Figure 5C), which is consistent with previous studies [18,19].

To further declare the exact target of exosomal miR-21-5p in regulating macrophage polarization. We used the STRING to construct the network relevant to the lists of target genes and DEGs, and the hub genes were identified in Cytoscape. The results indicated that SRC, a tyrosine kinase involved in macrophage-related inflammatory responses, is most likely a key hub gene in macrophage polarization. SRC is related to tumor suppressor factors, monocyte-macrophages and T cell chemokines by interacting with IL-15, colony stimulating factor 1 (CSF1), C-C motif chemokine ligand 5 (CCL-5), etc. (Figure 5D). Meanwhile, we found that RhoB is closely related to SRC and may be a potential target for mir-21-5p in macrophage M2 polarization.

To confirm our prediction, we transfected miR-21-5p mimics into THP-1 cells and examined expression of RhoB using Western-blot analysis. Comparing with a control group, we observed that RhoB’s level was significantly downregulated after transfection (Figure 6B). To determine whether miR-21-5p targets RhoB directly, we performed a reporter assay. The wild type and the mutated binding site of miR-21-5p of RhoB were cloned into luciferase vectors (Figure 6A). The result revealed that luciferase activity decreased markedly in THP-1 cells, carrying the wild type binding site vector, in the presence of miR-21-5p. However, cells containing the mutated binding site vector did not show such repression (Figure 6C). These results reveal that RhoB is a direct target of miR-21-5p in M2-like polarization of macrophages.

### 2.5. Inhibition of HCC-Derived Exosomal miR-21-5p Limits M2 Polarization of Macrophages in Response to Tumorous Education

To understand the function of RhoB in macrophage polarization, we knocked down RhoB with small interfering RNAs (siRNAs) in THP-1 cells, and the effect was identified using immunoblotting analysis. As shown in Figure 6D, the expression of RhoB was inhibited in THP-1 cells by treatment with siRNAs. Moreover, we found that the levels of SRC were decreased. We also detected the activation of the MAPK pathway, which is involved in the regulation of IL-10 production. Phosphorylation of MAPK signaling in macrophages is considered important for tumor-promoting cytokine production, and macrophage migration [20]. Therefore, we detected the phosphorylation of an extracellular signal-regulated kinase (Erk). The result revealed that the activation of Erk signaling upregulated in response to the suppression of RhoB. These data suggested that RhoB is a direct downstream target of miR-21-5p in regulating macrophage polarization.

To further confirm that HCC-derived exosomal miR-21-5p is the predominant effector in macrophage M2 polarization in TME, we designed a miR-21-5p sponge to suppress the levels of miR-21-5p. A miR-21-5p sponge sequence was constructed in lentivector. The construction virus was produced by transfecting lentiviral transfer and packaging plasmids into 293T cells, and the supernatant was harvested to treat THP-1 cells for 48 h before they were seeded on upper transwells. Meanwhile, HCC tumor cells were seeded on 24well plates. Subsequently, the two kinds of cells were co-cultured for 48 h. The effects on THP-1 cells were identified by immunoblotting analysis. To confirm the function of the sponge, we detected the expression level of B-cell lymphoma-2 (Bcl-2) and phosphatase and tensin homolog (PTEN), which were reportedly the targets of miR-21-5p in previous studies [21,22]. The results showed that downregulation of miR-21-5p in hepatic tumor cells limited the decrease in RhoB in macrophages. Similar results were also observed in the miR-21-5p sponge group (Figure 7B,C). It indicated that HCC-derived exosomal miR-21-5p played a key role in promoting M2-like polarization of macrophages in TME. Blocking exosomal miR-21-5p signaling will help attenuate the education of TME to macrophages and slow the malignant progression of HCC.

## 3. Discussion

In our study, we have shown that a large majority of HCC cases are with unfavorable prognosis in the background of accumulated M2-like TAM infiltration. TAMs are the most abundant immune cells in tumor tissues and M2-like macrophages are the predominant subtype. TAMs have been proved to exhibit significant immunosuppressive effects and promote tumor growth, angiogenesis, invasion, and metastasis by acting as a driver of M2-polarized macrophages [23]. It is commonly believed that TAMs are mainly derived from circulating monocytes. Blood monocytes are recruited into the tissue where they differentiate into macrophages or dendritic cells. However, recent research has showed that up to 50% of TAMs originate from resident macrophages in mouse models. Recent studies have proven that TAMs derived from hematopoietic stem cells (HSCs) are involved in immunosuppression and antigen presentation, while embryo-derived TAMs are responsible for angiogenesis in TME [24,25,26]. However, a growing number of studies show that TAM phenotypes are not fixed and that their dynamic alterations are implicated in hepatocarcinogenesis and its progression. The polarization of macrophages into M1 or M2 macrophages depends on the microenvironments [4,27,28].

In recent years, exosomes have been implicated in the regulation of this process. To certify our hypothesis, we treated macrophages with tumor-derived exosomes in vitro. We found that tumor-derived exosomes lead to the up-regulation of mir-21-5p in macrophages and promoted M2-like macrophage cytokines production. miR-21-5p, an anti-apoptotic regulator [29], is widely present in HCC tissues and is closely associated with poor clinical prognosis [30]. Previous studies report that miR-21-5p promotes the differentiation of myeloid monocytes into immunosuppressive-like macrophages [31]. In our study, we noted that modulating the level of miR-21-5p will affect the macrophage polarization. Emerging evidence supports the key roles of miRNA in macrophage polarization during HCC pathogenesis, including miR155, miR-149-5p, TUC339, miR-125a, etc. [32,33,34,35]. In addition, Li et al. found that exosomal miR-21-5p secreted by lung stromal cells could inhibit M1 polarization of alveolar macrophages and down-regulate the expression levels of IL-8, IL-1β, IL-6 and TNF-α, reducing the inflammatory response in the lungs [36]. In the study of Sahraei et al., they downregulated the expression of miR-21 in TAMs by carrier peptides, and found that tumor growth was inhibited and the anti-tumor immune response was reactivated [37]. 

We then predicted the target genes of miRNAs using bioinformatics and demonstrate that mir-21-5p regulates the MAPK signaling pathway in macrophages by targeting RhoB. RhoB is a member of the Rho subfamily of small GTPases, which switch the molecular cycle between an inactive GDP-bound form and active GTP-bound form [38]. RhoB has a high turnover rate, and it is a key regulator of diverse cellular processes, including the response to epidermal growth factor TGFβ, SRC activation [39]. Previous studies have shown that RhoB can affect macrophage morphology, adhesion, and migration by reducing the expression of β2 and β3 integrins on cell surfaces [40]. This may be one of the reasons for the massive infiltration of TAMs in tumor tissues. Recently, increasing evidence suggests that RhoB plays a role in the immune and inflammatory responses. One group of researches demonstrated that, upon LPS stimulation, the expression of RhoB increased and regulated toll-like receptor (TLR) activation via binding to major histocompatibility complex class II (MHCII) in macrophages, but they did not demonstrate whether RhoB affected MHCII expression at endosomal membranes [41]. In future, we will detect the relationship between RhoB and MHC II in macrophages, and this could give new insights into the underlying mechanism of RhoB in regulating antigen presentation.

In conclusion, our study demonstrated that RhoB serves as a potent target of HCC-derived miR-21-5p signaling pathways by interacting with Erk in M2-like macrophages and enhancing the production of pro-tumorous cytokines. Therefore, targeting M2 macrophages and intercepting their associated signaling pathways is a potential specific and novel therapeutic approach for HCC treatment.

## 4. Materials and Methods

### 4.1. Antibodies and Other Reagents

Antibodies against human HSP 70, β-Actin, Tubulin, and GAPDH were from Santa Cruz Biotechnology (Dallas, TX, USA). The TSG 101 and RhoB antibodies were from Protein-tech (Wuhan, Hubei, China). The SRC, Bcl-2, PTEN (138G6), Erk, and pErk (Thr202/Tyr204) antibodies were from Cell Signaling Technology (Danvers, MA, USA). Horseradish peroxidase (HRP)-conjugated secondary antibodies were from CWBIO (Changping, Beijing, China). Protease inhibitor cocktail was from Thermo Scientific (Waltham, MA, USA). PMA was from Sigma-Aldrich (St. Louis, MO, USA). PHK67 was from Umibio-tech (Shanghai, China). 

### 4.2. Cell Culture

The human THP-1 monocytic cell line was provided by Procell Life Science & Technology Co, Ltd. (Wuhan, Hubei, China). The human hepatocellular carcinoma cell lines HepG2, Hep3B, and HuH7 were obtained from American Type Culture Collection (ATCC, Manassas, VA, USA). THP-1 cells were cultured in RPMI-1640 medium supplemented with 10% FBS, 1% penicillin/streptomycin, and 0.05mM β-mercaptoethanol. Hepatocellular carcinoma cell lines were cultured in DMEM medium supplemented with 10% FBS and 1% penicillin/streptomycin. THP-1 cells (1 × 10^6^) were incubated with 200 ng/mL PMA for 48 h to induce them into macrophages. These cells were maintained in an incubator with 5% CO_2_ at 37 °C.

### 4.3. Western Blot

Cells were lysed on ice with RIPA buffer and protease and phosphatase inhibitor for 20 min. Lysate was centrifuged at 13,000 rpm for 15 min at 4 °C and the supernatant was transferred to a fresh tube. Protein concentration was qualified by a BCA Protein Assay kit (CWBIO, Beijing, China). Approximately 10 mg of protein per sample was loaded onto 12% SDS–PAGE gels and then transferred onto nitrocellulose membrane. The membrane was blocked with 5% non-fat dry milk in Tris-buffered saline for 1 h, and then incubated with primary antibodies at 4 °C for overnight, following by incubation with the horseradish peroxidase–conjugated secondary antibody at room temperature. Finally, the membranes were visualized with COMPLEXTM 2000.

### 4.4. qPCR

mRNA: Total RNA was extracted from cell lines using TRIzol reagent (CWBIO, Beijing, China) and reverse transcribed into cDNA using a Hifair^®^ III 1st Strand cDNA Synthesis Kit (YEASEN, Shanghai, China). Real-time PCR was performed using NovoStart^®^ SYBR High-Sensitivity qPCR SuperMix (Novoprotein Scientific Inc., Suzhou, Jiangsu, China). All of the primers were purchased from Tsingke Biotechnology Co., Ltd. (Beijing, China). The primers for amplification were: 

TNF-α, forward: 5′-CCTCTCTCTAATCAGCCCTCTG-3′, 

reverse: 5′-GAGGACCTGGGAGTAGATGAG-3′; 

IL-1β, forward: 5′-ATGATGGCTTATTACAGTGGCAA-3′, 

reverse: 5′-GTCGGAGATTCGTAGCTGGA-3′; 

IL-10, forward: 5′-GACTTTAAGGGTTACCTGGGTTG-3′, 

reverse: 5′-TCACATGCGCCTTGATGTCTG-3′; 

TGF-β1: forward: 5′-CTAATGGTGGAAACCCACAACG-3′, 

reverse: 5′-TATCGCCAGGAATTGTTGCTG-3′;

GAPDH: forward: 5′-GGAGCGAGATCCCTCCAAAAT-3′, 

reverse: 5′-GGCTGTTGTCATACTTCTCATGG-3′.

CCL4: forward: 5′-CTGTGCTGATCCCAGTGAATC-3′

reverse: 5′-TCAGTTCAGTTCCAGGTCATACA-3′.

Quantitative PCR was performed with Bio-Rad C1000 Thermal Cycler. Each amplification reaction was checked for the absence of nonspecific PCR products using melting curve analysis. The threshold cycle numbers obtained from qPCR were compared to generate the relative copy number as described by Livak and Schmittgen (2001) [42]. Data were normalized against GAPDH.

miRNA: After total RNA extraction, a tailing reaction was performed using *E. coli* Poly(A) Polymerase (NEB# M0276, Ipswich, MA, USA) and incubated at 37 °C for 30 min. Reverse transcription was performed using miRNA random primers. Real-time PCR was performed like mRNA. Data were normalized against U6. The primers for amplification were:

mir-21-5p: forward: 5′-GCAGTAGCTTATCAGACTGATG-3′,

reverse: 5′-GGTCCAGTTTTTTTTTTTTTTTCAAC-3′;

U6: forward: 5′-CTCGCTTCGGCAGCACATA-3′,

reverse: 5′-AACGATTCACGAATTTGCGT-3′.

### 4.5. RNA Interference

The miR-21-5p mimics (Gene Pharma, Suzhou, Jiangsu, China) were transfected into macrophages with Lipofectamine 2000 reagent (Invitrogen, Grand Island, NY, USA) at a final concentration of 40 nM according to the manufacturer’s instructions. The miR-21-5p mimic sequences were: sense: 5′-UAGCUUAUCAGACUGAUGUUGA-3′; antisense: 5′-AACAUCAGUCUGAUAAGCUAUU-3′. After transfection for 48 h, we collected whole-cell lysates for Western Blot or qPCR. 

### 4.6. Cell Viability Assay

Cells were seeded in 24-well plates at a density of 2 × 10^4^ cells per well in 500 μL. The medium was changed to a medium of macrophages followed by miRNA transfection after cells were attached. Cell viability was assessed by Cell Counting Kit-8 (CCK-8) (APExBIO, Shanghai, China) after 1 day or 2 days. 

### 4.7. Luciferase Reporter Assay

THP-1 cells (1 × 10^4^) were seeded into 24-well plates and transiently transfected with the different expression vectors together with complex of reporter plasmids and pCMV-β-galactosidase (β-Gal) using Lipofectamine 2000 (Invitrogen, Grand Island, NY, USA). Twenty-four hours after transfection, the cells were transfected with mir-21-5p mimics. Cells were then harvested and lysed on ice. Centrifuged supernatant (10,000 r.p.m, 10 min) was used to measure luciferase according to the manufacturer’s protocol (Promega, Madison, MI, USA).

### 4.8. Exosome Isolation and Identification

The exosomes were separated from the cell culture supernatant by ultrafiltration concentration and an ultracentrifugation method. To eliminate the effect of debris on the isolation of exosomes, the samples including cell culture supernatant were centrifuged at 300× *g* for 10 min, then 2000× *g* for 10 min. The supernatant were filtered with a 0.22 μm filter, then concentrated with 100 kDa ultrafiltration tube (Millipore, Billerica, MA, USA). The collected supernatants were centrifuged at 100,000× *g* for 70 min by ultra-centrifuge (Beckman, Indianapolis, IN, USA) twice. The precipitate was resuspended by particle-free PBS. After that, transmission electron microscopy was used to visualize the appearance of exosomes at a proportional scale of 0.2 μm. Furthermore, the exosomes’ size was detected using a Mastersizer 3000 (Malvern, Worcs, UK). The protein content of exosomes was qualified by a BCA Protein Assay kit (CWBIO, Beijing, China).

### 4.9. Analysis of HCC-Derived miRNA in Public Datasets

Data of HCC cell-derived exosomal miRNA were downloaded from the database GSE106452. We listed the top 30 highly expressed miRNAs by heatmap. TCGA LIHC data were analyzed by DESeq2 to identify differentially expressed miRNA genes between tumor and normal tissues in HCC (log_2_ FC > 1, FDR < 0.05). We determined the miRNAs that were highly expressed in both datasets by combining the results.

### 4.10. Analysis of miR-21-5p Expression and Survival Curve in HCC

The TCGA LIHC data were used to test the correlation of miR-21-5p and patient survival. The gene expression data and the clinical data were downloaded from UCSC Xena “http://xena.ucsc.edu/ (accessed on 12 September 2022)”. The differentiations of miR-21-5p expression have been shown by violin plot in normal and primary tumor tissues (Welch’s *t*-test, *p* = 2.064 × 10^−26^, t = −16.39). The primary tumor tissue samples were grouped into high and low expression groups by the mean value (log_2_ [RPM + 1] = 17.85). Kaplan–Meier survival curves were plotted to show differences in survival time (log-rank test statistics = 5.352, *p* = 0.02).

### 4.11. Enrichment Analysis of DEGs

Functional enrichment analysis of DEGs was performed by DAVID (The Database for Annotation, Visualization and Integrated Discovery) to identify GO (Gene Ontology) annotation. The data were download from GES 66805 and GES 95405. The results of DEGs were showed by heatmaps (log_2_ FC > 1, FDR < 0.05), and scatter plots was drawn using ggplot2 packages.

### 4.12. STRING Analysis

The protein–protein interaction (PPI) network of the target genes of mir-21-5p and DEGs was constructed using the online database STRING “https://string-db.org/ (accessed on 12 September 2022)”, and the functional interactions between proteins were analyzed. The combined score ≥ 0.400 is considered significant. We used Cytoscape to analyze hub genes, which are important nodes for visualization of PPI networks with many interactions.

### 4.13. Statistics Analysis

Data analysis was performed using the Graphpad Prism software version 7. Each experiment was carried out in triplicate, at least, and all results were presented as mean ± s.d. χ^2^-Test and Student’s *t*-test were used to assess statistical significance. A value of *p* < 0.05 was considered significant.

## Figures and Tables

**Figure 1 ijms-24-04593-f001:**
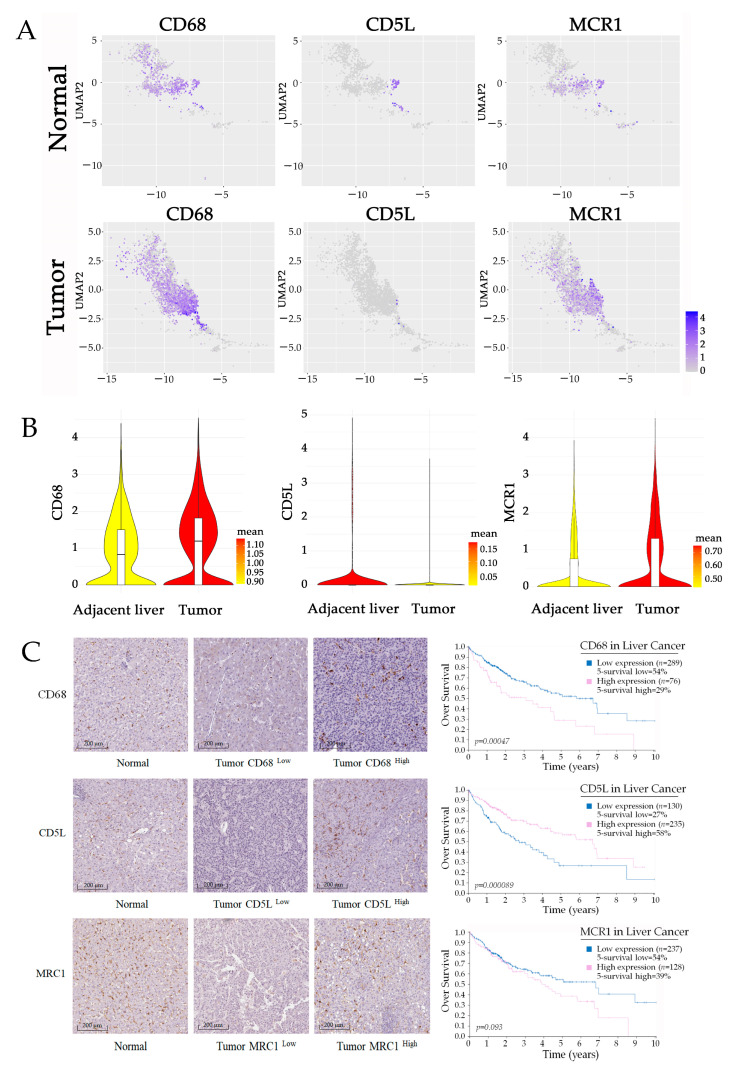
M2-like TAMs are associated with an unfavorable prognosis of HCC. (**A**,**B**) UMAP plot showing the expression levels of characteristic genes (CD68, CD5L, and MRC1) in macrophages in tumor and normal tissues. Violin plots indicating the expression of selected genes in macrophages from HCC samples. Data are from the online tool accessed at: “http://cancer-pku.cn:3838/HCC/ (accessed on 25 August 2022)”. (**C**) IHC double staining characterizes the distribution of macrophages in HCC, and Kaplan–Merier survival curves present the relationship between infiltration of TAM and prognosis of patients. The data are from The Human Protein Atlas “https://www.proteinatlas.org/ (accessed on 25 August 2022)”.

**Figure 2 ijms-24-04593-f002:**
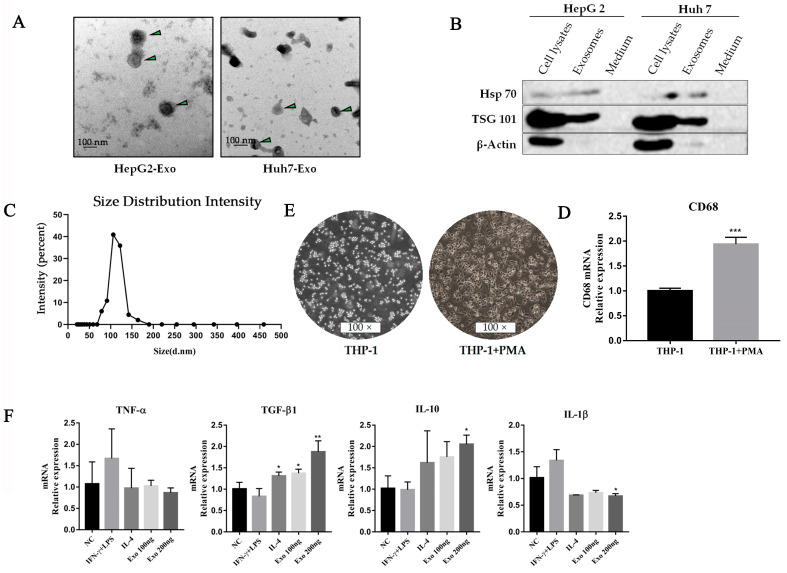
HCC-derived exosomes induced macrophage M2-like polarization. (**A**) Exosomes (HepG2 and Huh7) derived from HCC cell lines were collected by an ultrafiltration concentration and ultracentrifugation method. The morphology of exosomes was observed by transmission electron microscopy. The exosomes were marked with arrows. Scare bar: 100 nm. (**B**) Western blot analysis was used to detect exosome biomarkers (HepG2 and Huh7) derived from HCC cell lines, including Hsp70 and TSG101. (**C**) The sizes of HepG2 cell-derived exosomes were detected using a Mastersizer 3000. (**D**) Morphology characteristics of THP-1 cells after PMA treatment. THP-1 cells were seeded in 6-wells by 1 × 10^5^ cells per well and treated with PMA for 48 h. (**E**) qPCR was used to detect CD68 levels in THP-1 cells after PMA treatment, *** *p* < 0.001. (**F**) Cytokines were detected by qPCR after HepG2-derived exosomes treated macrophages for 48 h. ** *p* < 0.01, * *p* < 0.05.

**Figure 3 ijms-24-04593-f003:**
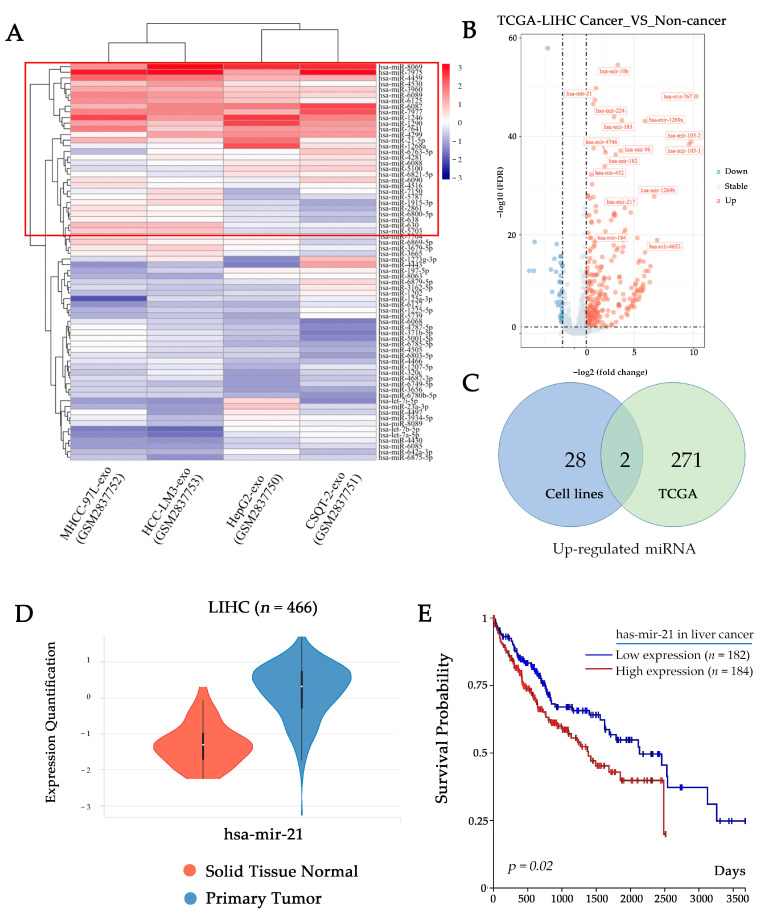
HCC-derived exosomes contain large amounts of miR-21-5p. (**A**) Heatmap shows top 30 miRNA (in the red box) in HCC cell line-derived exosomes (GSE106452). (**B**) Volcano plot shows differentially expressed miRNAs in TCGA LIHC dataset. (**C**) Venn diagram shows the intersection of up-regulated miRNA in GES106452 and TCGA LIHC datasets. (**D**) Violin plots show the expression of mir-21-5p in different types of tissue in TCGA LIHC. (**E**) Overall survival curves of TCGA LIHC data indicate the relationship between miRNA expression level and patient prognosis.

**Figure 4 ijms-24-04593-f004:**
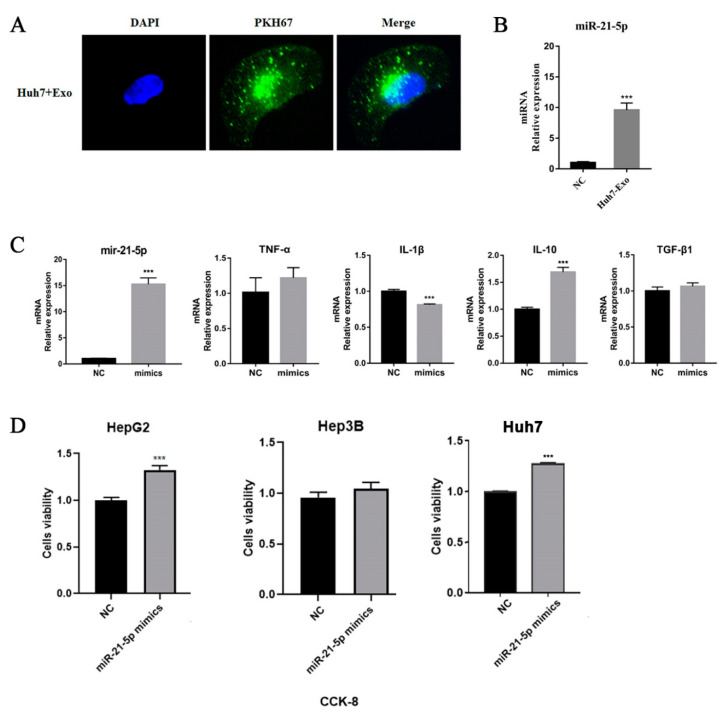
Tumor-derived exosomes regulate macrophage M2 polarization. (**A**) HCC cell line-derived exosomes were labeled by PKH67 and co-cultured with THP-1 cells for 24 h. The results were observed by fluorescence microscopy. (**B**) miR-21-5p level was detected by qPCR. (**C**) Transcriptional level of cytokines of THP-1 cells after mir-21-5p mimics transfection. 1 × 10^5^ THP-1 cells were seeded in 6-well plates and were treated with PMA for 48 h. miR-21-5p mimics were transfected into THP-1 cells using lipofectamine 2000. RNA was extracted from cells to perform qPCR detection. (**D**) The supernatant of THP-1 was used to treat HCC tumor cells (HepG2, Hep3B, and Huh7) for 48 h; cell viability was detected using CKK-8 assay. *** *p* < 0.001.

**Figure 5 ijms-24-04593-f005:**
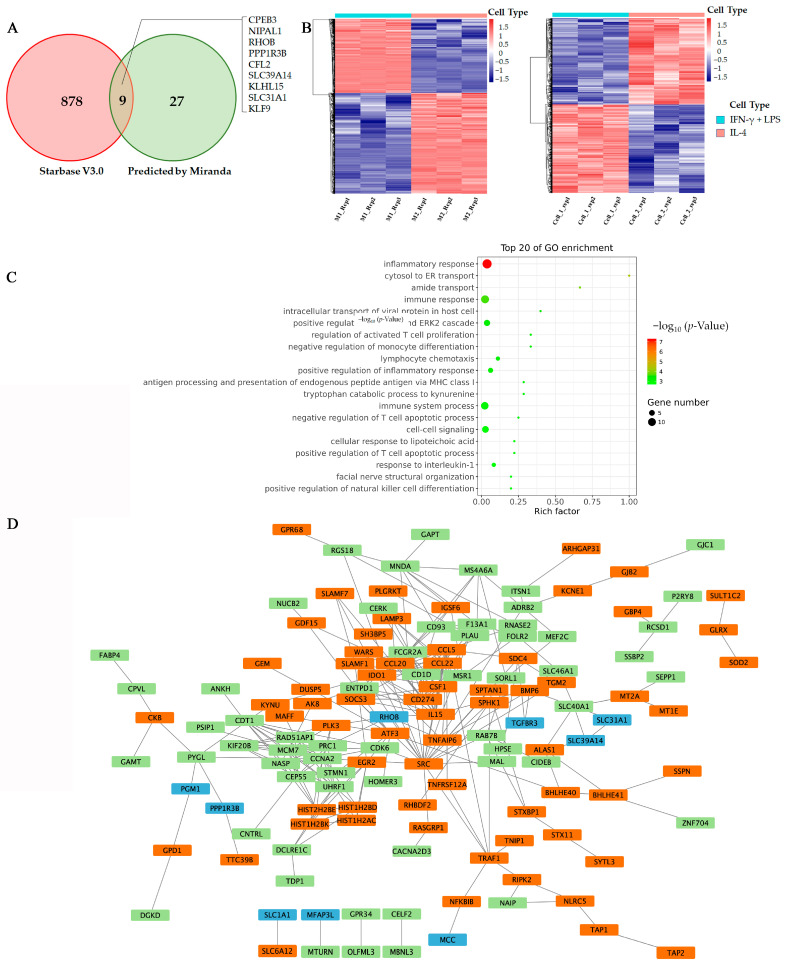
RhoB is a potential target for miR-21-5p. (**A**) Venn diagram shows the predicted targets of miR-21-5p, which were made by two kinds of bioinformatic tools. (**B**) Heatmap showing the DEGs between M1- and M2-polarized macrophages. Data were from GES 66805 and GES 95405. (**C**) Enrichment analysis of integrated DEGs. The color represents the *p*-value of the terms, while the *x*-axis represents different gene categories. (**D**) The network was constructed with the target genes of miR-21-5p and DEGs in macrophage polarization using the Search Tool for the Retrieval of Interacting Genes/Proteins (STRING). Blue boxes are the predicted targets of mir-21-5p, and green boxes are up-regulated genes, while the orange boxes are down-regulated genes during macrophage M2 polarization.

**Figure 6 ijms-24-04593-f006:**
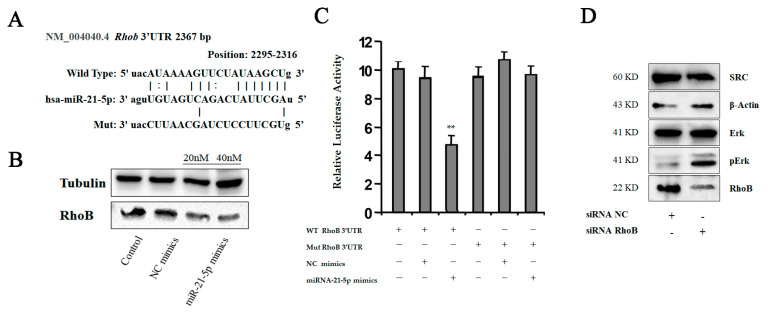
RhoB is the direct target for miR-21-5p in polarization of macrophages. (**A**) Sequence diagrams showing wild type and mutant type RhoB 3′UTR for miR-21-5p. (**B**) We transfected miR-21-5p mimics into THP-1 cells using lipofectamine 2000 and detected RhoB level using Western-blot analysis. (**C**) Reporter assay: THP-1 cells were transfected with wild type or mutant type RhoB 3′UTR vectors as indicated. After 24 h, the cells were infected with miRNA mimics. 14 h later, luciferase activity of reporters was measured, ** *p* < 0.05. (**D**) We transfected RhoB siRNA into THP-1 cells, and detected protein levels using Western-blot analysis as indicated.

**Figure 7 ijms-24-04593-f007:**
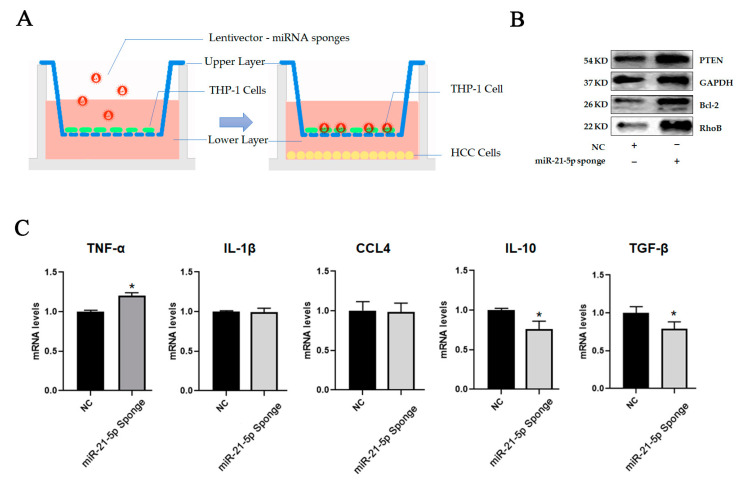
HCC-derived exosomal miR-21-5p regulated macrophage polarization by target RhoB. (**A**) THP-1 cells were seeded on upper layer after transfection with miRNA sponges. HCC cells were seeded on lower layer. Two kinds of cells were cultured for 48 h. (**B**) The levels of targets of miR-21-5p in HCC cells were detected using Western-blot analysis. (**C**) Cytokines in THP-1 were detected by qPCR, * *p* < 0.05.

## Data Availability

Data available upon reasonable request from corresponding author.

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
