# Peer review of "Hepatocellular Carcinoma Cell-Derived Exosomal miR-21-5p Induces Macrophage M2 Polarization by Targeting RhoB"

_ijms, 2023, doi:10.3390/ijms24054593_

Round 1

Reviewer 1 Report

The work of Haiyang Yu and colleagues presents interesting results elucidating the possible mechanism of how cancer cells interact with immune system cells forming microenvironment favorable for tumor progression. The study is well designed and logically consistent. In my opinion, it provides new facts of high scientific significance.   

Minor issues

-          Fig. 1. The description of the sub-images (A, B, C) in the legend does not correspond to the real situation.

-          I recommend revision of the Figures’ legends. They are not informative in their present state. The description must be focused on the results presented, but not only on approaches used in the study. Also, several images (for example, Figs. 1C, 3A, 3B, 5D, etc.) are blind or too small to distinguish details and text labels. Thus, their appearance should be improved.

-          Detailed criteria and parameters (if applicable) for every bioinformatics analysis must be included in the manuscript.

-          Please check using acronyms throughout the text. Not all of them are deciphered at the first mention.

-          Line 116. Results on CD63 are not presented in the manuscript.

-          Fig. 2E. These are data of western blot analysis; label ‘mRMA’ must be deleted from the axis. Also, please specify which cells were used as a source of exosomes for Fig. 2C, F.

Author Response

Dear reviewer:

    Thank you for your comments concerning our manuscript. Those comments are all valuable and helpful for revising and improving our paper, as well as the important guiding significance to our researches. We have studied comments carefully and have tried our best to improve and made some changes in the manuscript, which we hope meet with approval.  Revision notes, point-to-point, are given as follows:

-1  Fig. 1. The description of the sub-images (A, B, C) in the legend does not correspond to the real situation.

Response: We are sorry for the mistake. We have revised the figure legend and described it correctly in manuscript.

-2  The figure legends are not informative in their present state. Several images (for example, Figs. 1C, 3A, 3B, 5D, etc.) are blind or too small to distinguish details and text labels.

Response: We have revised the figure legends and adjusted resolution of figures in manuscript.

-3  Detailed criteria and parameters (if applicable) for every bioinformatics analysis must be included in manuscript.

Response: We have added the details of bioinformatics analysis in "Materials and Methods" section. 

-4  Please check using acronyms throughout the text. Not all of them are deciphered at the first mention.

Response: We have standardized the use of abbreviations, and made corrections in manuscript.

-5  Line 116. Results on CD63 are not presented in manuscript.

Response: Sorry for the mistake, we did not detect CD63 level in exosomes. We have corrected the mistake in manuscript.

-6  Fig. 2E. These are data of western blot analysis; label ‘mRNA’ must be deleted from the axis. Also, please specify which cells were used as a source of exosomes for Fig. 2C, F.  

Response: We are sorry for the mistake in figure legend. The transcription level of CD68 was detected by qPCR, we have corrected it in manuscript. Also, we have figured out the source of exosomes for Fig. 2C, F in figure legend.

Reviewer 2 Report

The article titled "Hepatocellular Carcinoma Cells-derived Exosomal miR-21-5p Induces Macrophages M2 Polarization by Targeting RHOB" authored by Haiyang Yu and colleagues investigates the role of exosomes derived from hepatocellular carcinoma (HCC) cells and suggest that they are mediating the differentiation of tumor-associated macrophages (TAMs) into a suppressive, M2-like phenotype. The study demonstrates that HCC-derived exosomes promote the differentiation of macrophages into M2-like macrophages through the regulation of miR-21-5p. This miRNA directly targets RhoB, disrupting MAPK signaling pathways and contributes to the malignant growth of HCC cells.

The study appears to be well designed and executed, and the results are presented in a clear and organized manner. The conclusion that targeting M2-like TAMs and their associated signaling pathways could provide a therapeutic approach for HCC treatment is an intersting statement based on the available presented data.

Overall, this study highlights the complex interplay between cancer cells and the surrounding microenvironment. Further research is needed to confirm and extend these findings, but the study presents a promising direction for future investigations into the treatment of HCC.

minor revisions

Some grammatical and spelling errors exist:

eg. line 28 axle -> axis

line 306 approaches -> approach etc.

Author Response

Dear reviewer:

    Thank you for your comments concerning our manuscript. Those comments are all valuable and helpful for revising and improving our paper, as well as the important guiding significance to our researches. We have studied comments carefully and have tried our best to improve and made some changes in the manuscript, which we hope meet with approval.  Revision notes, point-to-point, are given as follows:

-1  Some grammatical and spelling errors exist:  eg. line 28 axle -> axis, line 306 approaches -> approach etc.

Response: We have checked the grammar and spelling carefully and corrected the mistakes in manuscript.
